# OpenReview forum: "On the Challenges of Using Black-Box APIs for Toxicity Evaluation in Research"
_EMNLP/2023/Conference — EMNLP 2023 Main_

### Official Review · Reviewer_5s8T · 2023-07-21

**Typos Grammar Style And Presentation Improvements:** 1. Line 061
**Soundness:** 4

**Excitement:**

4: Strong: This paper deepens the understanding of some phenomenon or lowers the barriers to an existing research direction.

**Missing References:**

Wu et al. (2023) - "Fine-Grained Human Feedback Gives Better Rewards for Language Model Training" may be of interest. They use Perspective as a reward model for feedback learning.

**Paper Topic And Main Contributions:**

Black-box APIs like Perspective are widely used to detect toxic content, including toxic content generated by language models. This paper highlights the problems that are caused by this practice, especially regarding the reproducibility and instability of research results due to (irregular and often unannounced) updates to the API. To back up their arguments, the authors present clear experiments that show major changes in high-profile benchmarking sets for large language models.

**Questions For The Authors:**

1. Have you considered arguing more against the use of black-box APIs in general? In your recommendations for living benchmarks, for example, would the use of an open-access model not be a good alternative to Perspective? Such a model will have its own issues and limitations, but it will be open and static.
2. Similarly, have you considered scoring models / evaluating RealToxicityPrompts with some open-access alternative to Perspective and seeing how this affects results?
3. You mostly focus on generative language models. What about applications of Perspective in classification settings and other analyses?
4. Have you considered drawing parallels between using Perspective API in research and using commercial generative model APIs like GPT, Claude etc. in research? Seems like many of the same arguments apply.


**Reasons To Accept:**

1. The authors identify an intuitive but critical problem with using black-box toxicity detection APIs for research. They provide strong evidence to back up their claims.
2. Since Perspective in particular is so widely used, their findings are relevant to a lot of NLP research.
3. The recommendations made by the authors based on their findings are sound and should be widely adopted.

Overall, the paper presents a good point and makes it well. I strongly believe it should be accepted.


**Reasons To Reject:**

The framing of the work suggests a more general analysis, but the actual experiments all focus on a single API – Perspective. The authors could be clearer and more open about this, and it would not make their work less impactful.

**Reproducibility:**

5: Could easily reproduce the results.

**Reviewer Confidence:**

5: Positive that my evaluation is correct. I read the paper very carefully and I am very familiar with related work.

---

> ### Author Rebuttal · Authors · 2023-08-28
>
> We extend our gratitude to the thought-provoking questions raised by the reviewer **5s8T**. We also thank their recognition of work as rigorous “provide strong evidence to back up their claims” and widely applicable “findings are relevant to a lot of NLP research.”
>
>
> > **The framing of the work suggests a more general analysis, but the actual experiments all focus on a single API – Perspective. The authors could be clearer and more open about this, and it would not make their work less impactful.**
>
> Thank you for your feedback on the framing of the work. We chose to frame the paper more generally as Perspective API is the most widely used toxicity detection tool in research. We’ll adjust the wording to make sure readers understand our focus on the Perspective API scores from the start.
>
>
> > **Have you considered arguing more against the use of black-box APIs in general? In your recommendations for living benchmarks, for example, would the use of an open-access model not be a good alternative to Perspective? Such a model will have its own issues and limitations, but it will be open and static.**
>
> We believe that open-access models would be a good alternative to Perspective, because updates would be visible to the public. However, there’s not currently **an open-source model similar in quality to Perspective**. There are some open-sourced models that were used in other studies [1, 2], but they have some flaws in comparison to Perspective such as more limited language and domains. Also, for toxicity and hate-speech studies, **we believe that an ever-improving model is beneficial**. As we understand, those topics are not static over time.
> We argue that the main problem of using Perspective for research today is the lack of API version traceability, as you pointed out in your suggestion. This is something that could be solved by API maintainers or at least tractable amongst the community by open-sourcing continuations.
>
> [1]​​ Unitary AI’s “Detoxify”
>
> [2] Antypas, Dimosthenis, and Jose Camacho-Collados. "Robust Hate Speech Detection in Social Media: A Cross-Dataset Empirical Evaluation." arXiv preprint arXiv:2307.01680 (2023).
>
>
> > **Similarly, have you considered scoring models / evaluating RealToxicityPrompts with some open-access alternative to Perspective and seeing how this affects results?**
>
> No, we have not considered scoring with different open-access alternatives. The main objective of our work is to show distributional changes for black-box APIs, in particular Perspective API. To achieve this, it is essential to possess pre-existing scores for comparison. The majority of available open-sourced continuations and scores are for the Perspective API and RealToxicityPrompts pair.
> Additionally, from previous work on design biases of toxicity datasets and models [3], we expect results from other models to be comparable only at a high level to those of Perspective API, given the profound influence of dataset annotators on the results.
>
> [3] Santy, Sebastin, et al. "NLPositionality: Characterizing Design Biases of Datasets and Models." arXiv preprint arXiv:2306.01943 (2023).
>
>
> > **You mostly focus on generative language models. What about applications of Perspective in classification settings and other analyses?**
>
> Assessing the impact of Perspective API in other settings such as classification would be of great value to the work. As elucidated in Section 3.1, we expect to have even more intense distributional changes for the other attributes returned from the API as toxicity is amongst the three attributes that changed the least out of the 8 according to Wasserstein distances. However, **we do require previously open-sourced scores to make that assessment**, and the other attributes from Perspective are used less often than toxicity in research.
>
>
> > **What is the meaning of the colours of the connecting lines in Figure 1?**
>
> In Figure 1, colors are defined by the alphabetical order of model names. In this sense, from HELM’s nomenclature convention, models from the same organization should have similar colors (i.e. AlephAlpha models are red, Cohere models are orange, and OpenAI’s are blue.)

---

### Official Review · Reviewer_Lwgv · 2023-08-05

**Soundness:** 4

**Excitement:**

4: Strong: This paper deepens the understanding of some phenomenon or lowers the barriers to an existing research direction.

**Paper Topic And Main Contributions:**

This paper studies the issue of using blackbox APIs to assess a model's toxicity as blackbox APIs are continuously updated while the published results remain static. The authors show a rigorous analysis comparing published and rescored results for a wide variety of models for the RealToxicityPrompts dataset and UDDIA. The authors outline a series of brief recommendations for stakeholders to better navigate this line of research in terms of reproducibility.

**Questions For The Authors:**

The last two paragraphs of Section 3.2 are difficult to understand. Are you trying to say that the published results, which combine prompts and continuations to score for toxicity, are incorrect and that they should be scored separately only for the continuations? If so, don't the diagrams indicate that the models are more toxic than they actually are since when scoring only the continuations lead to a lower score? It seems lines 272-278 are saying the opposite.

**Reasons To Accept:**

- The paper highlights an important issue and its impact of using blackbox APIs that get continuously updated.
- The authors perform a rigorous study on a wide set of models and toxicity datasets.
- The authors present recommendations for each stakeholder in toxicity evaluation research.

**Reasons To Reject:**

The recommendations are relatively shallow compared to the performed analysis. It would have been more interesting to suggest a systematic setup that can help practitioners easily employ the provided recommendations, but this is beyond the scope of this paper.

**Reproducibility:**

4: Could mostly reproduce the results, but there may be some variation because of sample variance or minor variations in their interpretation of the protocol or method.

**Reviewer Confidence:**

4: Quite sure. I tried to check the important points carefully. It's unlikely, though conceivable, that I missed something that should affect my ratings.

**Typos Grammar Style And Presentation Improvements:**

- Figure 3: Since the x-axis is not related serially (at least for the last two, generations rescored vs prompts and generations rescored), I would suggest the plots be drawn with bar graphs rather than a line graph. It's fine for Figure 4, consider making the same change as well for consistency.
- Line 277 they are would be -> they would be
- Figure 1 is only mentioned in page 5 but it appears in the paper much earlier. It would be better to mention Figure 1 earlier, even without going into detail, to help with the flow or consider moving Figure 1 closer to where the first mention appears.
- Table 3 does not add too much information on top of Figure 1 and may be more suitable to be placed in the appendix.

---

> ### Author Rebuttal · Authors · 2023-08-28
>
> We thank reviewer **Lwgv** for their positive and detailed feedback, acknowledging we show “a rigorous analysis comparing published and rescored results for a wide variety of models” and how we outline “recommendations for stakeholders to better navigate this line of research in terms of reproducibility”. Additionally, we fixed the typos and grammar mistakes.
>
>
> > **The recommendations are relatively shallow compared to the performed analysis. It would have been more interesting to suggest a systematic setup that can help practitioners easily employ the provided recommendations, but this is beyond the scope of this paper.**
>
> We appreciate the reviewer's recognition of the paper's limited scope regarding providing a systematic setup. Our plan involves sharing all the code we used for analysis. This code will assist other researchers in both scoring continuations with Perspective API and evaluating models using standardized metrics, in line with the suggestions from Section 4. This step will particularly benefit those who adopt our key recommendation: ensuring consistent comparisons by rescoring previous continuations. While our ability to promote the most impactful recommendation (explicit model versioning) is constrained, we have engaged with well-known toxicity scoring APIs to advocate for improved transparency in versioning practices.
>
>
> > **The last two paragraphs of Section 3.2 are difficult to understand. Are you trying to say that the published results, which combine prompts and continuations to score for toxicity, are incorrect and that they should be scored separately only for the continuations?**
>
> Thanks to the reviewer for this opportunity to clarify. We clarify that prompt and continuations are typically always scored separately for toxicity. However, **the standard practice is that authors use old prompt scores (inherited from prior work) and only rescore the fresh continuation scores (scenario 2)**. This is what we point to as _incorrect_. It is incorrect when the versions of toxicity scores conflict between prompts and completions because these were scored using different versions of the API. Scenario 3 (technically correct) is what the results would be if both prompts and continuations had fresh scores (i.e. if authors scored prompts and continuations **separately, but under the same API version**). We are committed to making this more clear for the reader in the final manuscript, thanks again to the reviewer for their feedback.
>
>
> > **If so, don't the diagrams indicate that the models are more toxic than they actually are since when scoring only the continuations lead to a lower score? It seems lines 272-278 are saying the opposite.**
>
> Yes, the diagrams indicate that. When using fresh scores only for the continuations (scenario 2), models appear to be less toxic than they really are (scenario 3). We’ll update our manuscript so this is more clear.

---

### Official Review · Reviewer_U24C · 2023-08-05

**Soundness:** 4

**Excitement:**

3: Ambivalent: It has merits (e.g., it reports state-of-the-art results, the idea is nice), but there are key weaknesses (e.g., it describes incremental work), and it can significantly benefit from another round of revision. However, I won't object to accepting it if my co-reviewers champion it.

**Paper Topic And Main Contributions:**

This paper revolves around the evolving perception of toxicity and the continuous retraining of commercially available APIs like the Perspective API for toxicity detection. The study evaluates the impact of these changes on the reproducibility of research that compares various models and methods designed to address toxicity.

**Reasons To Accept:**

1. The paper pointed out an important problem of using Black-Box APIs for Toxicity Evaluation. This is an important finding because authors’ joint usage of outdated and fresh scores prevents a fair comparison of different techniques over time and leads authors to biased conclusions.
2. Supported by detailed data analysis and illustrations, the author clearly presents how  REALTOXICITYPROMPTS distribution changes over time. And presented the impact of API changes on rankings of model risk, living benchmarks, and reproducibility of research contributions.
3. Authors provided several good recommendations on resolving the problem to both API maintainers and authors.


**Reasons To Reject:**

1. The paper raises a significant concern regarding the limitations of black-box APIs used for toxicity evaluation. However, its focus is relatively narrow, centering on a specific research domain. As a result, the potential audience for this paper could be limited.
2. The author's major concern is the data distribution drift of the black-box API poses a challenge to the reproducibility of scientific results. However, the reviewer thinks a dynamic always up-to-date API may have more advantages. First, no evaluation dataset is perfect. Having an API that’s evolving overtime is definitely a good thing for the community and provides more accurate evaluation results. Second, many existing static benchmarks or evaluation datasets are already abused by researchers. Many later studies are overfitting on these static benchmarks and evaluation datasets. That being said, the recommendations from the author are still valid. The authors and API providers can do better in this field.


**Reproducibility:**

3: Could reproduce the results with some difficulty. The settings of parameters are underspecified or subjectively determined; the training/evaluation data are not widely available.

**Reviewer Confidence:**

4: Quite sure. I tried to check the important points carefully. It's unlikely, though conceivable, that I missed something that should affect my ratings.

---

> ### Author Rebuttal · Authors · 2023-08-28
>
> We thank reviewer **U24C** for their invaluable feedback and for acknowledging the importance of the contribution and rigor of our “detailed analysis” showing the impact black-box API scores have on rankings of model risk and reproducibility of research results, which “leads authors to biased conclusions” that jeopardize reproducibility.
>
>
> > **The paper raises a significant concern regarding the limitations of black-box APIs used for toxicity evaluation. However, its focus is relatively narrow, centering on a specific research domain. As a result, the potential audience for this paper could be limited.**
>
> We understand how the focus of the paper could seem narrow at first, as we exclusively evaluate how Perspective API’s scores have changed over time. _However, similar reproducibility difficulties are true for **any** black-box API that does not inform of model updates or provides model versioning for users._ Nowadays, only a handful of enterprises and groups have access to the amount of computing necessary to train the most powerful LLMs, and users have access to those exclusively through an API. Similar to the difficulties we found when using Perspective, previous work has shown the lack of reproducibility in general use text generation APIs [1, 2]. We believe these work, in conjunction with ours, to be of extreme importance for setting clear limitations (and room for improvement) for the usage of machine learning algorithms through APIs. We’ll make sure to add this parallel in the paper so that our possible audience is increased.
>
> [1] Ruis, Laura, et al. "Large language models are not zero-shot communicators." arXiv preprint arXiv:2210.14986 (2022).
>
> [2] Chen, Lingjiao, Matei Zaharia, and James Zou. "How is ChatGPT's behavior changing over time?." arXiv preprint arXiv:2307.09009 (2023).
>
>
> > **The author's major concern is the data distribution drift of the black-box API poses a challenge to the reproducibility of scientific results. However, the reviewer thinks a dynamic always up-to-date API may have more advantages. First, no evaluation dataset is perfect. Having an API that’s evolving overtime is definitely a good thing for the community and provides more accurate evaluation results. Second, many existing static benchmarks or evaluation datasets are already abused by researchers. Many later studies are overfitting on these static benchmarks and evaluation datasets. That being said, the recommendations from the author are still valid. The authors and API providers can do better in this field.**
>
> We thank the reviewer for acknowledging that our recommendations are valid and that both authors and API providers can do better. We agree with the reviewer that having an API that’s evolving over time is a good thing for the community. However, **our main concern is not about data drift itself or dataset overfit**, but about the **structural problem of lack of explicit model and API versioning** in ML-based platforms. As APIs are being increasingly used in research, we expect such practices to exacerbate the risk of data drift and reproducibility difficulties [1, 2]. Our most effective recommendation in this regard would be to make model updates transparent to users. This would allow for responsible science in which we are sure apples-to-apples comparisons are being made.

---

### Meta-Review · Area_Chair_61FS · 2023-09-16

**Recommendation:** 5

**Metareview:**

**Summary:**
The paper describes the impact of continuous updates to blackbox APIs for toxicity detention, often commercially available, like the Perspective API, on research reproducibility. The authors conduct a rigorous analysis comparing  different models and methods aimed at addressing toxicity, They emphasize issues related to the instability in research result due to irregular and frequently unannounced API updates. The contribution reveals major changes in high-profile benchmarking datasets for large language models such as RealToxicityPrompts dataset and UDDIA. Furthermore, the authors provide a set of concise recommendations for stakeholders to better navigate this line of research in terms of reproducibility.

**Strengths:**
The reviewers unanimously agree on the paper's strengths, which are consistent across all three reviews. Here, we'll highlight the points initially outlined by the first review, which also apply to the other two reviews.

- The paper pointed out an important problem of using Black-Box APIs for Toxicity Evaluation. This is an important finding because authors’ joint usage of outdated and fresh scores prevents a fair comparison of different techniques over time and leads authors to biased conclusions.
- Supported by detailed data analysis and illustrations, the author clearly presents how REALTOXICITYPROMPTS distribution changes over time. And presented the impact of API changes on rankings of model risk, living benchmarks, and reproducibility of research contributions.
- Authors provided several good recommendations on resolving the problem to both API maintainers and authors.

**Weaknesses:**
Reviewers are in consensus concerning weaknesses. In particular, the actual experiments all focus on a single API – Perspective. As a result, the potential audience for this paper could be limited. Additionally, the first reviewer emphasizes that having an API that’s evolving overtime is definitely a good thing for the community and provides more accurate evaluation results. Furthermore, numerous existing static benchmarks or evaluation datasets are already overused by researchers. Consequently, this resulting in overfitting on these static benchmarks and evaluation datasets in subsequent studies.

**Author-Reviewer discussion and acknowledgment:**
The authors have provided clarifications regarding the reviewers' concerns and have outlined how the paper will be enhanced in response. All the reviewers have taken into account the arguments and points emphasized by the authors in their rebuttal.

**Conclusion:**
The paper is well-structured, but the reviewers recommend that the authors include additional references. Furthermore, the reviewers suggest that the authors carefully review the paper to enhance the clarity of the English presentation and rectify the identified typos.

---

### Decision · Program_Chairs · 2023-10-07

**Decision:**

Accept-Main

**Comment:**

**Summary:**
The paper describes the impact of continuous updates to blackbox APIs for toxicity detention, often commercially available, like the Perspective API, on research reproducibility. The authors conduct a rigorous analysis comparing  different models and methods aimed at addressing toxicity, They emphasize issues related to the instability in research result due to irregular and frequently unannounced API updates. The contribution reveals major changes in high-profile benchmarking datasets for large language models such as RealToxicityPrompts dataset and UDDIA. Furthermore, the authors provide a set of concise recommendations for stakeholders to better navigate this line of research in terms of reproducibility.

**Strengths:**
The reviewers unanimously agree on the paper's strengths, which are consistent across all three reviews. Here, we'll highlight the points initially outlined by the first review, which also apply to the other two reviews.

- The paper pointed out an important problem of using Black-Box APIs for Toxicity Evaluation. This is an important finding because authors’ joint usage of outdated and fresh scores prevents a fair comparison of different techniques over time and leads authors to biased conclusions.
- Supported by detailed data analysis and illustrations, the author clearly presents how REALTOXICITYPROMPTS distribution changes over time. And presented the impact of API changes on rankings of model risk, living benchmarks, and reproducibility of research contributions.
- Authors provided several good recommendations on resolving the problem to both API maintainers and authors.

**Weaknesses:**
Reviewers are in consensus concerning weaknesses. In particular, the actual experiments all focus on a single API – Perspective. As a result, the potential audience for this paper could be limited. Additionally, the first reviewer emphasizes that having an API that’s evolving overtime is definitely a good thing for the community and provides more accurate evaluation results. Furthermore, numerous existing static benchmarks or evaluation datasets are already overused by researchers. Consequently, this resulting in overfitting on these static benchmarks and evaluation datasets in subsequent studies.

**Author-Reviewer discussion and acknowledgment:**
The authors have provided clarifications regarding the reviewers' concerns and have outlined how the paper will be enhanced in response. All the reviewers have taken into account the arguments and points emphasized by the authors in their rebuttal.

**Conclusion:**
The paper is well-structured, but the reviewers recommend that the authors include additional references. Furthermore, the reviewers suggest that the authors carefully review the paper to enhance the clarity of the English presentation and rectify the identified typos.